# Verified, Shared, Modular, and Provenance Based Research Communication with the Dat Protocol

**Chris Hartgerink** 

Independent Researcher, 10115 Berlin, Germany; chris@libscie.org

**Abstract:** A scholarly communication system needs to register, distribute, certify, archive, and incentivize knowledge production. The current article-based system technically fulfills these functions, but suboptimally. I propose a module-based communication infrastructure that attempts to take a wider view of these functions and optimize the fulfillment of the five functions of scholarly communication. Scholarly modules are conceptualized as the constituent parts of a research process as determined by a researcher. These can be text, but also code, data, and any other relevant pieces of information that are produced in the research process. The chronology of these modules is registered by iteratively linking to each other, creating a provenance record of parent and child modules (and a network of modules). These scholarly modules are linked to scholarly profiles, creating a network of profiles, and a network of how profiles relate to their constituent modules. All these scholarly modules would be communicated on the new peer-to-peer Web protocol Dat, which provides a decentralized register that is immutable, facilitates greater content integrity than the current system through verification, and is open-by-design. Open-by-design would also allow diversity in the way content is consumed, discovered, and evaluated to arise. This initial proposal needs to be refined and developed further based on the technical developments of the Dat protocol, its implementations, and discussions within the scholarly community to evaluate the qualities claimed here. Nonetheless, a minimal prototype is available today, and this is technically feasible.

**Keywords:** metaresearch; decentralization; decentralization; publishing; p2p

## 1. Introduction

In scholarly research, communication needs to be thorough and parsimonious in logging the order of various research steps, while at the same time being functional in seeking and distributing knowledge. Roosendaal and Geurts proposed that any scholarly communication system needs to serve as a (1) registration, (2) certification, (3) awareness, and (4) archival system [1]. Sompel and colleagues added that it also needs to serve as an (5) incentive system [2].

How the functions of scholarly communication are conceptualized and implemented directly impacts (the effectiveness of) scholarly research. For example, an incentive system might be present where a number of publications or a publication outlet is more important than the quality of the publications [3]. In a narrow sense, this scholarly communication system serves the fifth function of providing an incentive system. In a wider sense, it undermines the goal of scholarly research, which scholarly communication is a part of, and therefore does not serve its purpose.

Narrow conceptualizations of the functions of a scholarly communication system can be identified throughout the current article-based system. Registration occurs for published works, but registration is incomplete due to selective publication (e.g., one out of two registered clinical trials gets published; [4]), making research highly inefficient [5]. Certification occurs through peer review [6], but peer review is confounded by a set of human biases at the reporting and evaluation stages (e.g., methods are evaluated as of higher quality when they result in statistically-significant

results than when they result in statistically nonsignificant results; [7]), leading to the "natural selection of bad science" [8]. Awareness occurs, but increasingly only for those researchers with the financial means to access or make it accessible [9]. Restrictions on the sharing of scholarly information hamper discovery and widespread dissemination. Content is archived, but is centralized (i.e., failure prone), separated from the main dissemination infrastructure, and not available until an arbitrary trigger event occurs (i.e., a dark archive; [10]).

The scholarly paper seems an anachronistic form of communication in light of how we now know it undermines the functions it is supposed to serve. When no alternative communication form was feasible (i.e., before the Internet and the web), the scholarly paper seemed a reasonable and balanced form for communication. However, already in 1998, seven years after the first web browser was released, researchers associated with the scholarly publisher Elsevier suggested to make changes to the way scholars communicate scholarly research [11]. More specifically, they suggested to change the communication to a more modular form, which would help iterate research more frequently and increase feedback moments (high speed of feedback was essential to, for example, Nature's rise during the early Twentieth Century; [12]). Throughout the years, others also suggested various perspectives on modularity [13,14] and suggested micro- and nano-publications [14,15]. One example of modular, stepwise research communication is depicted in Figure 1.

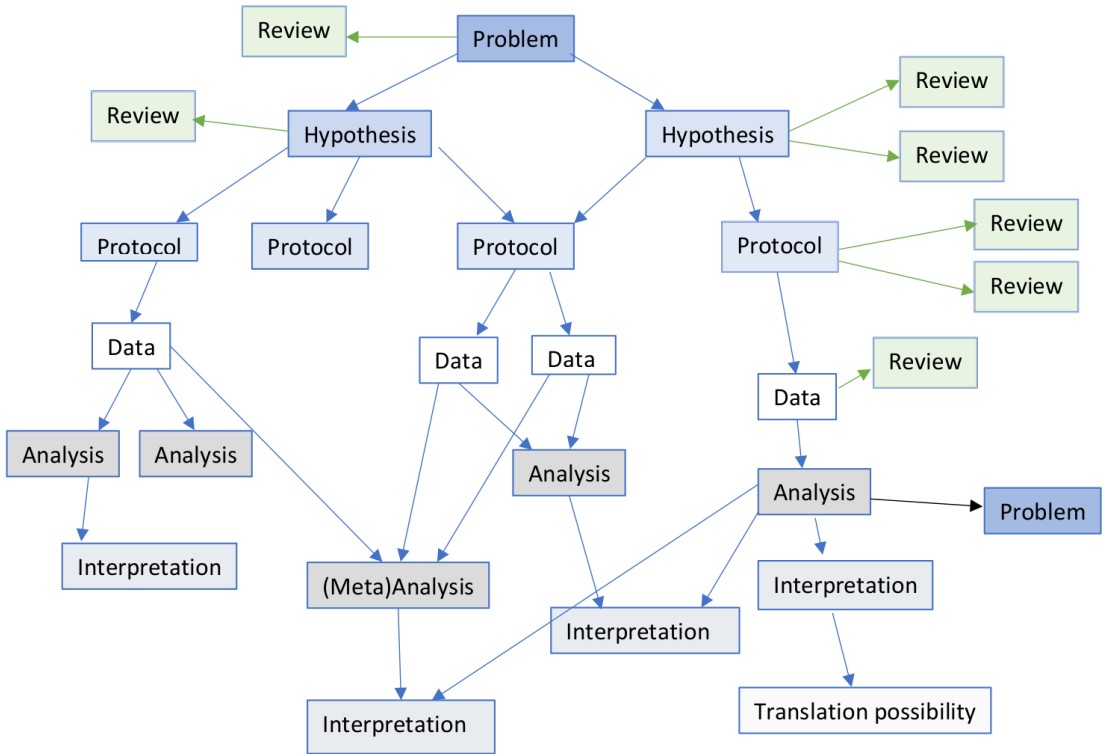

**Figure 1.** An example of modular, stepwise research communication, from the Octopus project (see also https://perma.cc/TA79-YPH9).

Modular scholarly outputs, each a separate step in the research process, could supplement the scholarly article (as detailed in [16]). Scholarly textbooks and monographs (i.e., vademecum science; [17]) communicate findings with few details and a high degree of certainty; scholarly articles present relatively more details and less certainty than textbooks, but still lack the detail to reproduce results. This lack of detail is multiplied by the increasingly complex research pipelines due to technological changes and the size of data processed. Moreover, textbooks and articles construct narratives across findings because they report far after events have happened, and this is what editors expect. Scholarly modules could serve as the base for scholarly articles, reporting more details, less

certainty of findings, and where events are reported closer to their occurrence. Granular reporting may help facilitate a shift from authorship to contributorship [18] and could facilitate reproducibility (i.e., it is easier to reproduce one action with more details than multiple actions with fewer details per action); earlier reporting could facilitate discussion by making it practical for the research process (extending the idea of Registered Reports; [19]) and making content easier to find and reuse. As findings become replicated and more consensus about a finding starts to arise, findings could move up the "chain" and be integrated into scholarly articles and textbooks. Articles and books would then provide overviews and larger narratives to understand historical developments within scholarly research. Figure 2 provides a conceptual depiction of how these different forms of documenting findings relate to each other.

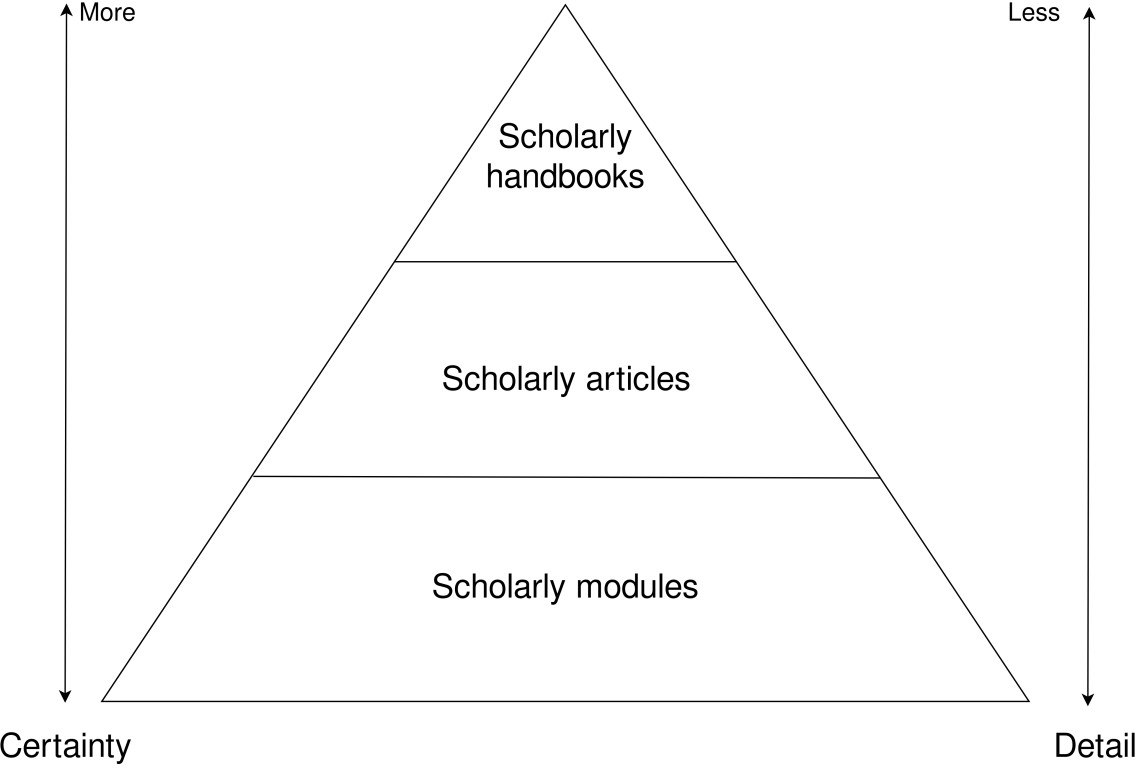

**Figure 2.** Conceptual depiction of how different forms of scholarly communication relate to each other in both detail and certainty.

Below, I extend on technical details for a modular scholarly communication infrastructure that facilitates (more) continuous communication and builds on recent advances in web infrastructures. The premise of this scholarly infrastructure is a wider interpretation of the five functions of a scholarly communication system, where: (1) registration is (more) complete; (2) certification by peer review is supplemented by embedding chronology to prevent misrepresentation and by increased potential for verification and peer discussion; (3) unrestricted awareness (i.e., access) is embedded in the underlying peer-to-peer protocol that locks it open-by-design; (4) archiving is facilitated by simplified copying; and (5) making more specific scholarly evaluation possible to improve incentives (for an initial proposal of such evaluation systems, see [16]). First, I expand on the functionality of the Internet protocol Dat (https://datproject.org) and how it facilitates improved dissemination and archiving. Second, I illustrate an initial design of modular scholarly communication using this protocol to facilitate better registration and certification.

## 2. Dat Protocol

The Dat protocol (`dat://`) is a peer-to-peer protocol, with persistent public keys per filesystem (see also https://perma.cc/FX7M-H85Y; [20,21]). Each filesystem is a folder that lives on the Dat network. Upon creation, each Dat filesystem receives a unique 64-character hash address, which provides read-only access to anyone who has knowledge of the hash. Below, an example filesystem is presented. Each Dat filesystem has a persistent public key, which is unaffected by bit-level changes within it (e.g., when a file is modified or created). Other peer-to-peer protocols, such as BitTorrent or the Inter Planetary File System (IPFS), receive new public keys upon bit-level changes in the filesystem and require re-sharing those keys after each change (at the protocol level).

```
0c6...613/
|--- file1
|--- file2
|--- file3
|--- file4
```

Bit-level changes within a Dat filesystem are verified with cryptographically-signed hashes of the changes in a Merkle tree. In effect, using a Merkle tree creates a verified append-only register. In a Merkle tree, contents are decomposed into chunks that are subsequently hashed in a tree (as illustrated in Figure 3), adding each new action to the tree at the lowest level. These hashes are cryptographically signed with the permitted users' private keys. The Dat protocol regards all actions in its filesystem as `put` or `del` commands to the filesystem, allowing all operations on the filesystem to be regarded as actions to append to a register (i.e., log). For example, if an empty `file5` was added to the Dat filesystem presented above, the register would include `[put] /file5 0 B (0 blocks)`; if we delete the file, it would log `[del] /file5`. The complete register for this Dat filesystem is as follows:

```
dat://0c6...613

1 [put] /file1 0 B (0 blocks)
2 [put] /file2 0 B (0 blocks)
3 [put] /file3 0 B (0 blocks)
4 [put] /file4 0 B (0 blocks)
5 [put] /file5 0 B (0 blocks)
6 [del] /file5
```

The persistent public key combined with the append-only register results in persistent versioned addresses for filesystems that also ensure content integrity. For example, based on the register presented above, we see that Version 5 includes `file5`, whereas Version 6 does not. By appending +5 to the public key (`dat://0c66...613+5`), we can view the Dat filesystem as it existed at Version 5 and be ensured that the content we received is the exact content for that version. If the specific Dat filesystem is available from at least one peer on the network, it means that both "link rot" and "content drift" [22,23] could become superfluous.

Any content posted to the Dat protocol is as publicly available as the public key of that Dat filesystem that is shared. More specifically, the Dat protocol is inherently open. As such, if that key is widely shared, the content will also be harder or impossible to remove from the network because other peers (can) have copied it. Conversely, if that key is shared among just a few people, that content can more easily disappear from the network, but will remain more private. This is important in light of privacy issues, because researchers cannot unshare personal data after they have widely broadcast it. However, because the Dat protocol is a peer-to-peer protocol and users connect directly to each other, information is not mediated. The protocol uses package encryption by default, which can also

help improve secure and private transfers of (sensitive) data. Users would (most likely) also remain personally responsible for the information they (wrongly) disclose on the network.

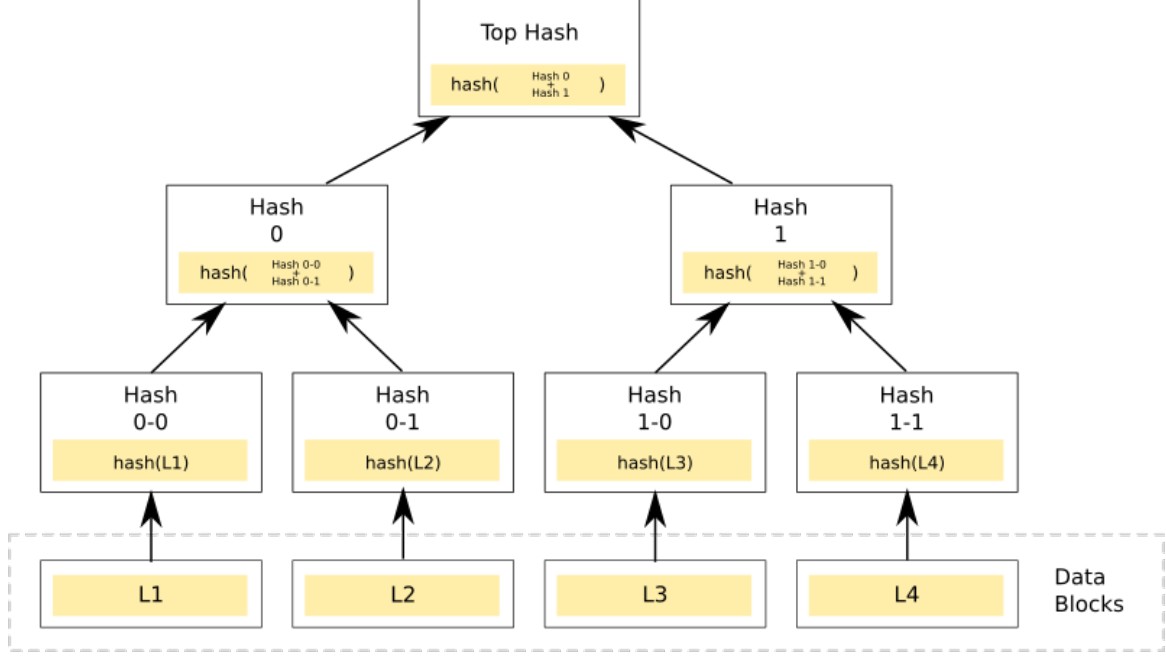

**Figure 3.** A diagram conceptually depicting how a Merkle tree hashes initial chunks of information into one top hash, with which the content can be verified. These do not correspond to the actions outlined in the text.

## 3. Verified Modular Scholarly Communication

Here, I propose an initial technical design of verified modular scholarly communication using the Dat protocol. Scholarly modules are instantiated as separate Dat filesystems for each researcher or for each module of scholarly content. Scholarly content could entail virtually anything the researcher wants or needs to communicate in order to verify findings (see also [16]). Hence, there is no restriction to text as there is in the current article-based scholarly communication system; it may also include photographs, data files, scripts, etc. Note that all presented hypothetical scenarios include shortened Dat links, and the unshortened links can be found in the Supplementary File S1.

### 3.1. Scholarly Profiles

Before communicating research modules, a researcher would need to have a place to broadcast that information. Increasingly, researchers are acquiring centralized scholarly profiles to identify the work they do, such as ORCIDs, ResearcherIDs, Google Scholar profiles, or ResearchGate profiles. A decentralized scholarly profile in a Dat filesystem is similar and provides a unique ID (i.e., public key) for each researcher. However, researchers can modify their profiles freely because they retain full ownership and control of their data (as opposed to centralized profiles) and are not tied to one platform. As such, with decentralized scholarly profiles on the Dat network, the researcher permits others access to their profiles instead of a service permitting them to have a profile.

Each Dat filesystem is initialized with a `dat.json` with some initial metadata, including its own Dat public key, the title (i.e., name) of the filesystem, and a description. For example, Alice wants to create a scholarly profile and initializes her Dat filesystem, resulting in:

```
{
"title": "Alice",
"description": "I am a physicist at CERN-LHC. As a fan of the
decentralized Web, I look forward to communicating my research in
a digital native manner and in a way that is not limited to just text."
text.",
"url": "dat://b49...551"
}
```

Because `dat.json` is a generic container for metadata across the Dat network, I propose adding `scholarly-metadata.json` with some more specific metadata (i.e., data about the profile) for a scholarly context. At the bare minimum, we initialize a scholarly profile metadata file as:

```
{
"type": "scholarly-profile",
"url": "dat://b49...551",
"parents": [],
"roots": [],
"main": "/cv.pdf",
"follows": [],
"modules": []
}
```

where the `type` property indicates it is a scholarly profile. The `url` property provides a reference to the public key of Alice herself (i.e., self-referencing). The `parents` property is where Alice can indicate her "scholarly parents" (e.g., supervisors, mentors); the `roots` property is inherited from her scholarly parents and links back to the root(s) of her scholarly genealogy. The `main` property indicates the main file for Alice's profile. The `follows` property links to other decentralized scholarly profiles or decentralized scholarly modules that Alice wants to watch for updates. Finally, the `modules` property refers to versioned scholarly modules, which serve as Alice's public registrations. These metadata files may be joined in a later specification.

Assuming Alice is the first person in her research program to use a decentralized scholarly profile, she is unable to indicate `parents` or inherit `roots`. However, Bob and Eve are her PhD students, and she helps them set up a decentralized scholarly profile. As such, their profiles do contain a parent: Alice's profile. Based on this genealogy, we would be able to construct self-reported genealogical trees automatically for scholarly profiles. Bob's `scholarly-metadata.json` subsequently looks as follows:

```
{
"type": "scholarly-profile",
"url": "dat://c3a...a1b",
"parents": [ "dat://b49...551" ],
"roots": [ "dat://b49...551" ],
"main": null,
"follows": [],
"modules": []
}
```

Alice wants to stay up to date with the work from Bob and Eve and adds their profiles to the `follows` property. By adding the unique Dat links to their scholarly profiles to her `follows` property, the profiles can be watched in order to build a chronological feed that continuously updates. Whenever Bob (or Eve) changes something in his profile, Alice gets a post in her chronological feed; for example,

when Bob follows someone, when Eve posts a new scholarly module, or when Bob updates his `main` property. In contrast to existing social media, Alice can either fully unfollow Bob, which removes all of Bob's updates from her feed, or "freeze follow", where she simply does not get any future updates. A "freeze follow" follows a static and specific version of the profile by adding a version number to the followed link (e.g., `dat://...+12`).

Using the `follows` property, Alice can propagate her feed deeper into her network, as depicted in Figure 4. More specifically, Alice's personal profile, rank zero in the network, extends to the people she follows (i.e., Bob and Eve are rank one). Subsequently, the profiles Bob and Eve follow are of rank three. By using recursive functions to crawl the extended network to rank *N*, edges in the network are easily discovered despite the (potential) lack of direct connections [24].

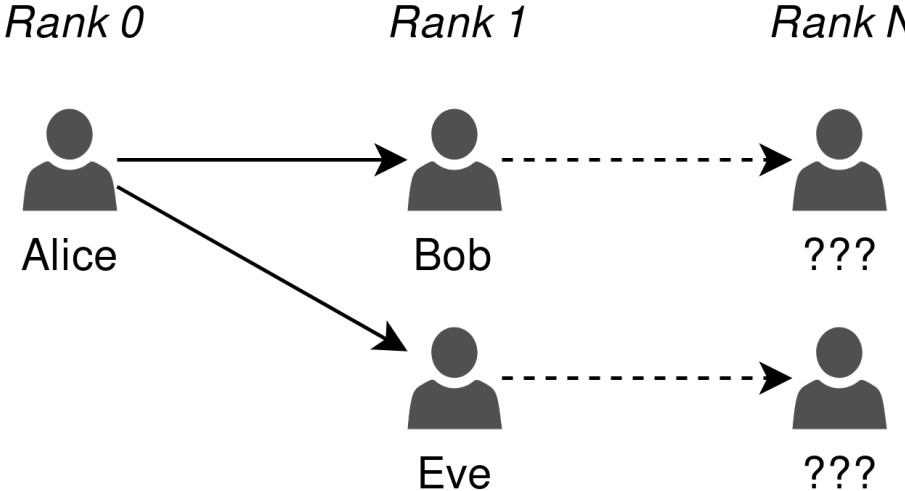

**Figure 4.** Conceptual diagram of scholarly profiles and following others. Network propagation to rank N can be used to facilitate the discovery of researchers and to build networks of researchers.

The `main` property can be used by a researcher to build a personalized profile beyond the metadata. For example, Alice wants to make sure that people who know the Dat link to her scholarly profile can access her curriculum vitae, so she adds `/cv.pdf` as the `main` to her scholarly profile. Whenever she submits a job application, she can link to her versioned scholarly profile (e.g., `dat://b49...551+13`). Afterwards, she can keep updating her profile whatever way she likes. She could even choose to host her website on the decentralized web by attaching a personal web page with `/index.html`. Because of the versioned link and the properties of the Dat protocol, she can rest assured that the version she submitted is the version the reviewing committee sees. Vice versa, whenever she receives a versioned link to a scholarly profile, she can rest assured it is what the researcher wanted her to see.

The `modules` property contains an array of versioned Dat links to scholarly modules. What these scholarly modules are and how they are shaped is explained in the next section. The `modules` property differs from the `follows` property in that it can only contain versioned Dat links, which serve as registrations of the outputs of the researcher. Where a versioned link in the `follows` property is regarded as a "freeze follow," a versioned link in the `modules` property is the registration and public communication of the output. The versioned links also prevent duplicate entries of outputs that are repeatedly updated. For example, a scholarly module containing a theory could be registered repeatedly over the timespan of several days or years. If the researcher registered non-versioned links of the scholarly module, registration would not be specific, and the scholarly profile could contain duplicates. By including only versioned links, the registrations are specific and unique.

### 3.2. Scholarly Modules

Scholarly research is composed of time-dependent pieces of information (i.e., modules) that chronologically follow each other. For example, predictions precede data and results, otherwise they become postdictions. In a typical theory-testing research study, which adheres to the framework of a modern empirical research cycle [25], we can identify at least eight chronological modules of research outputs: (1) theory, (2) predictions, (3) study design, (4) study materials, (5) data, (6) code for analysis, (7) results, (8) discussion, and (9) summary (these are examples, and modules should not be limited to these to prevent homogenization of scholarly outputs; [26]). Sometimes, we might iterate between steps, such as adjusting a theory due to insights gathered when formulating the predictions. Continuously communicating these in the form of modules as they are produced, by registering versioned references to Dat filesystems in a scholarly profile as explained before, could fulfill the five functions of a scholarly communication system and is unconstrained by the current article-based system (see also [16]).

These scholarly modules each live in their own filesystem, first on the researcher's computer and, when synchronized, on the Dat network. Hence, researchers can interact with files on their own machine as they are used to doing. The Dat network registers changes in the filesystem as soon as it is activated. As such, researchers can initialize a Dat filesystem on their computer and, for example, copy private information into the filesystem, anonymize it, and only then activate and synchronize it with the Dat network (note: this does not require connection to the Internet, but initialization of the protocol). The private information will then not be available in the version history of the Dat filesystem.

Metadata for scholarly modules also consists of a generic `dat.json` and a more specific `scholarly-metadata.json`. The `dat.json` contains the title of the module, the description, and its own Dat link. For example, Alice communicates the first module on the network, where she proposes a theory; the `dat.json` file for this module is:

```
{
"title": "Mock Theory",
"description": "This is a mock theory but it could just as well be
a real one.",
"url": "dat://dbf...d82"
}
```

Again, more specific metadata about the decentralized scholarly module is added in `scholarly-metadata.json`. At the bare minimum, the metadata for a scholarly module is initialized as:

```
{
"type": "scholarly-module",
"url": "dat://dbf...d82",
"authors": [
"dat://b49...551",
"dat://167...a26"
],
"parents": [],
"roots": [],
"main": "/theory.md"
}
```

These metadata indicate aspects that are essential in determining the content and provenance of the module. First, we specify that it is a scholarly module in the `type` property. Second, we specify its own Dat `url` for reference purposes. Third, an array of Dat links in the `authors` property links

to scholarly profiles for authorship. Subsequently, if the module is the following step of a previous registered module, we specify the Dat link of the preceding module(s) in the `parents` property in the form of a versioned Dat link. Tracing the parents' parents forms a chronology of findings, leading ultimately to the `roots` property. In practice, the `roots` property is inherited from the immediate parents. Because the presented hypothetical module above is the first on the network, it has no parents or roots. The `main` property specifies a single landing page/file of the scholarly module. For a text-based scholarly module, `main` might be `/index.html` (or `/theory.md` as it is here), whereas for a data module, that could be `/data.csv`. For more complex modules, a guidebook to navigate the module could be included. The researcher can also store other relevant assets in the Dat filesystem, such as converted files or supporting files. For the text-based scholarly module, assets could include figures; for data-based scholarly modules, assets could include codebooks.

To register a module into the researcher's profile, the versioned Dat link is included in the `modules` array on the profile. More specifically, when the registration process is initiated, the Dat filesystem is inspected for the latest version number, which is appended to the Dat link before it is put in the `modules` property. Specifically for Alice's theory, she was at Version 19 when she wanted to register it. This means that `dat://dbf...d82+19` is appended to the `modules` array in her scholarly profile. All the users who follow Alice get an update that she registered her theory, with a versioned link that is unique and persistent, referring to exactly the content Alice registered. Alice can keep updating her theory locally, without it affecting what the people who follow her see, because it does not affect Version 19. When the module is registered, others can view the most recent version of the Dat filesystem (e.g., theory) by removing the version from the Dat link (or view any other synchronized version if available from the network).

Figure 5 depicts how the scholarly modules relate to each other (Panel B). The versioned, registered scholarly modules become the parent and root links in subsequent child modules. For example, a set of predictions links back to the theory they are distilled from; a study design links back to the predictions it is planned to test and by extension to the theory on which it is based. Panel B in Figure 5 conceptually depicts one contained empirical research cycle registered in this way. The links between versioned scholarly modules embed the chronological nature of the research process in its communication.

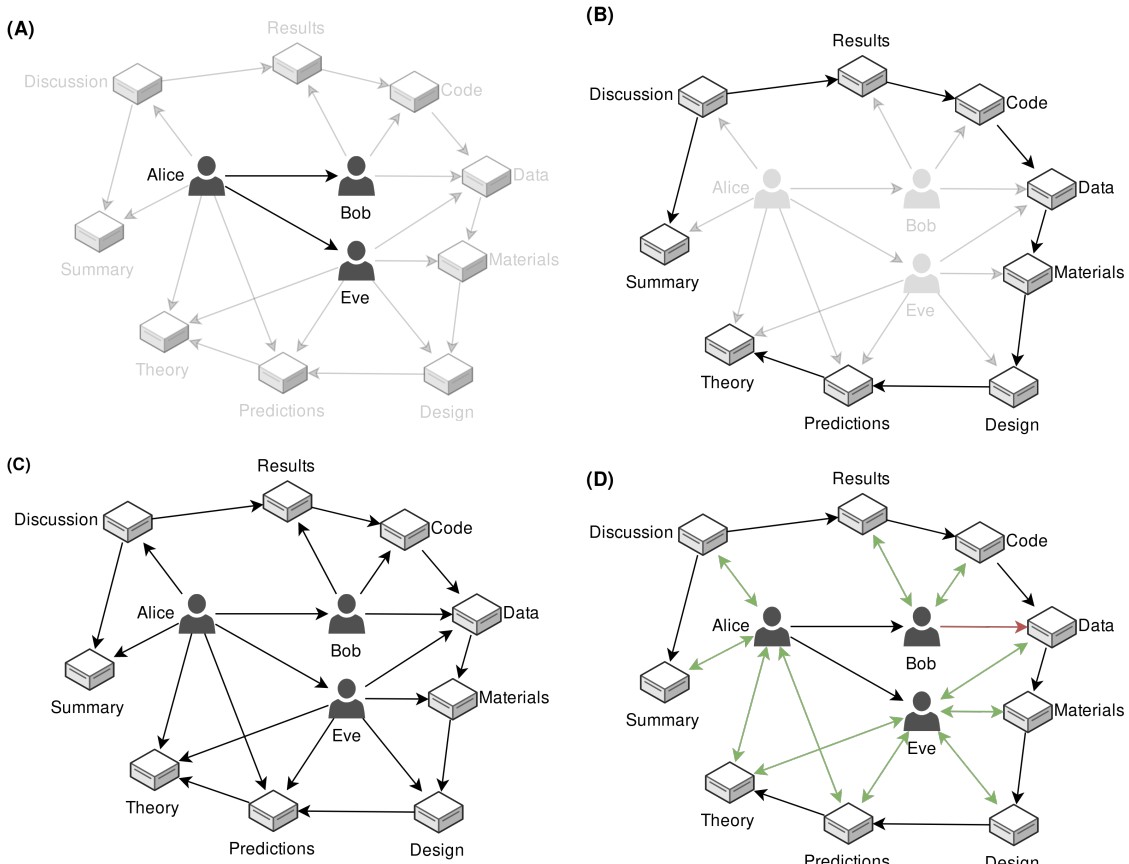

**Figure 5.** Conceptual representations of how scholarly profiles relate to each other (**A**), how scholarly modules relate to each other (**B**), how scholarly profiles and modules create a network of scholarly activity in both researchers and research (**C**), and how claims of authorship are verified if two-way or unverified if one-way (**D**).

*3.3. Verification*

In order to detect whether scholarly modules that a researcher claims to have authored are indeed (partly) theirs, the scholarly module needs to also assign the profile as the author. For example, Alice and Eve claim to have authored Version 19 of the "Theory" module in their profiles (Figure 5, Panel C). Because a module can only be edited by its author, we can inspect the scholarly module to corroborate this. For verified authorship, the module should ascribe authorship to Alice and Eve. To do this, we inspect `scholarly-metadata.json` of the "Theory" module for the registered version (i.e., Version 19). If the versioned theory module also ascribes authorship to Alice or Eve, we have two-way verification of authorship (Figure 5, Panel D). In other words, registered scholarly modules must corroborate the authorship claims of the scholarly profiles in order to become verified.

Unverified authorship can happen when a researcher incorrectly claims authorship over a module or when a module ascribes authorship to a researcher who does not claim it. In Figure 5, Panel D, for example, Bob has claimed authorship of the data module, which is not corroborated by the scholarly module. Unverified authorship of this kind (i.e., where a researcher incorrectly claims authorship) is helpful in preventing misrepresentation of previous work by that researcher. Unverified authorship where a researcher is incorrectly ascribed authorship can have various origins. A researcher might remove a versioned module from their profile, effectively distancing themselves from the module (similar to retracting the work, but on a more individual level). In a similar vein, it might also be that the author registered a later version of the module in their profile and deleted the old version (similar to a corrigendum). Note that the registration will still be available in the history of the profile, because the history of a Dat filesystem is append-only.

*3.4. Prototype*

In order to show that decentralized, modular scholarly communication is not just a hypothetical exercise, I developed a minimal working prototype. The prototype code supplied below currently only functions within the Beaker Browser because specific Application Programmatic Interfaces (APIs) that directly interface with the Dat protocol are not yet available in the most commonly-used web browsers (e.g., Mozilla Firefox, Google Chrome).

The minimal working prototype ingests a network of decentralized scholarly modules and profiles. More specifically, it ingests all content to rank *N* of the network, using `webdb` (https://github.com/beakerbrowser/webdb). `webdb` collects the scholarly metadata from each scholarly module and scholarly profile and consolidates these disparate pieces of information into a local database. This database can be considered temporary; the original information still has its primary origin in the disparate scholarly modules and scholarly profiles that live on the Dat network. As such, the same database can be reconstructed at any time without any issues, assuming the modules are still available. Figure 6 presents a screenshot of the prototype, which looks like any other web page to the user, but does not have a centralized server providing the content. Note also the link at the bottom showcasing the versioned link to the analysis file.

Procedurally, the prototype takes Alice's scholarly profile as the starting point, subsequently ingesting the network presented in Figure 5. By doing so, we get a one-on-one replication of Alice's perspective (regardless of whether we are Alice or not). As such, Alice's Dat link serves as the starting point (rank zero). The metadata contained in her profile are ingested into our local database. Subsequently, the links in her profile to other scholarly modules (or profiles) are ingested into the database (rank one), and the links they have (rank two), and so on (to rank *N*). The following JavaScript code produces this local database for Alice specifically (`dat://b49...551`), but can be replaced with Bob's, Eve's, or anyone else's scholarly profile to receive their personal network.

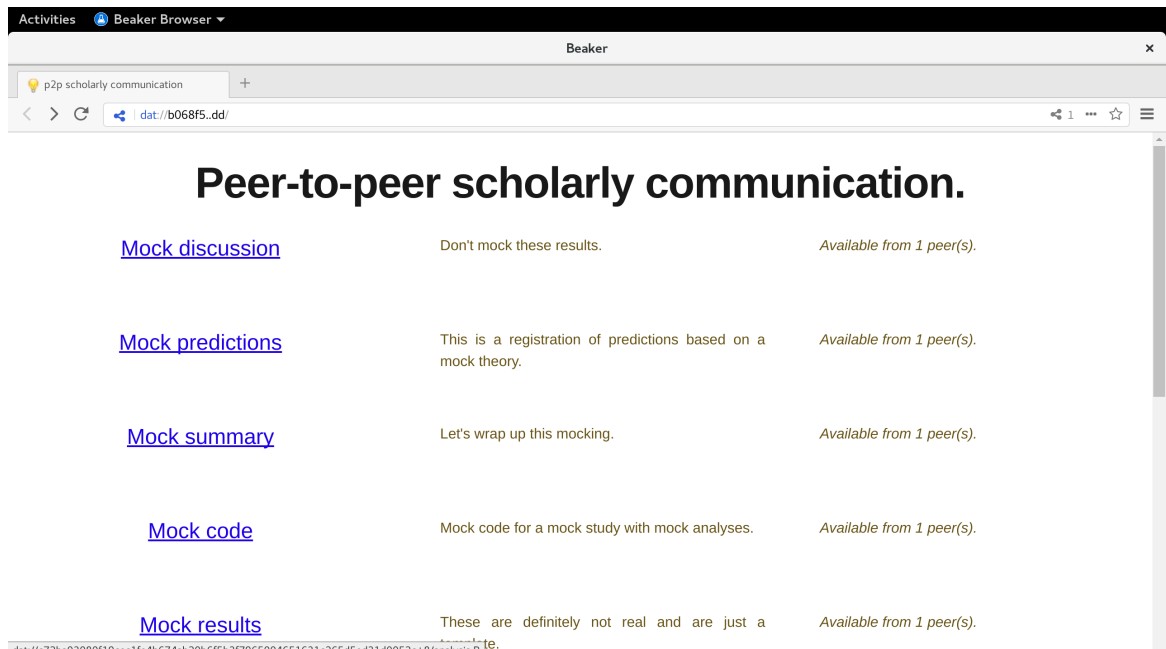

**Figure 6.** Screencap of the minimal prototype of decentralized scholarly communication. The prototype resembles a regular web page on the user side, but on the backend, it runs entirely on Dat filesystems that live on a decentralized network.

```javascript
// npm install -g @beaker/webdb
const WebDB = require('@beaker/webdb')

let webdb = new WebDB('view')

webdb.define('modules', {
filePattern: [ '/scholarly-metadata.json' ],
index: [ 'type', 'authors', 'parents', 'root',
'main', 'follows', 'modules' ]
})

async function ingestPortal (url) {
 await webdb.open()

let archive = new DatArchive(url)
 await webdb.indexArchive(url)

let scholRaw = await archive.readFile(
'/scholarly-metadata.json')

let scholParsed = await JSON.parse(
  scholRaw)

if (scholParsed.type === 'scholarly-profile') {
console.log(scholParsed)
scholParsed.follows.concat(
scholParsed.modules).forEach((val) => {
ingestPortal(val)
})
}
}

ingestPortal("dat://b49...551")
```

The presented prototype provides a portal to the information contained in the modules, but is not the sole portal to access that information. Because the modules live on a decentralized network and are open-by-design, anyone may build a portal to view that information (Figure 7 presents a mockup of an additional interface). As such, this is not a proposal for a platform, but for an infrastructure. The difference between platforms and infrastructure is vital in light of the ownership and responsibility of communicated content and the moderation of that content. As opposed to centralized services that carry the legal burden and therefore moderate its platform, this type of infrastructure does not take such a role and merely aims to facilitate the individual. As a consequence, the legal burden remains with the individual. Moreover, platforms require people to go to one place (e.g., you cannot view content of ResearchGate on Academia.edu or Elsevier's content on Wiley's web page); this infrastructure would give the potential for various types of usage to take place on the same type of infrastructure.

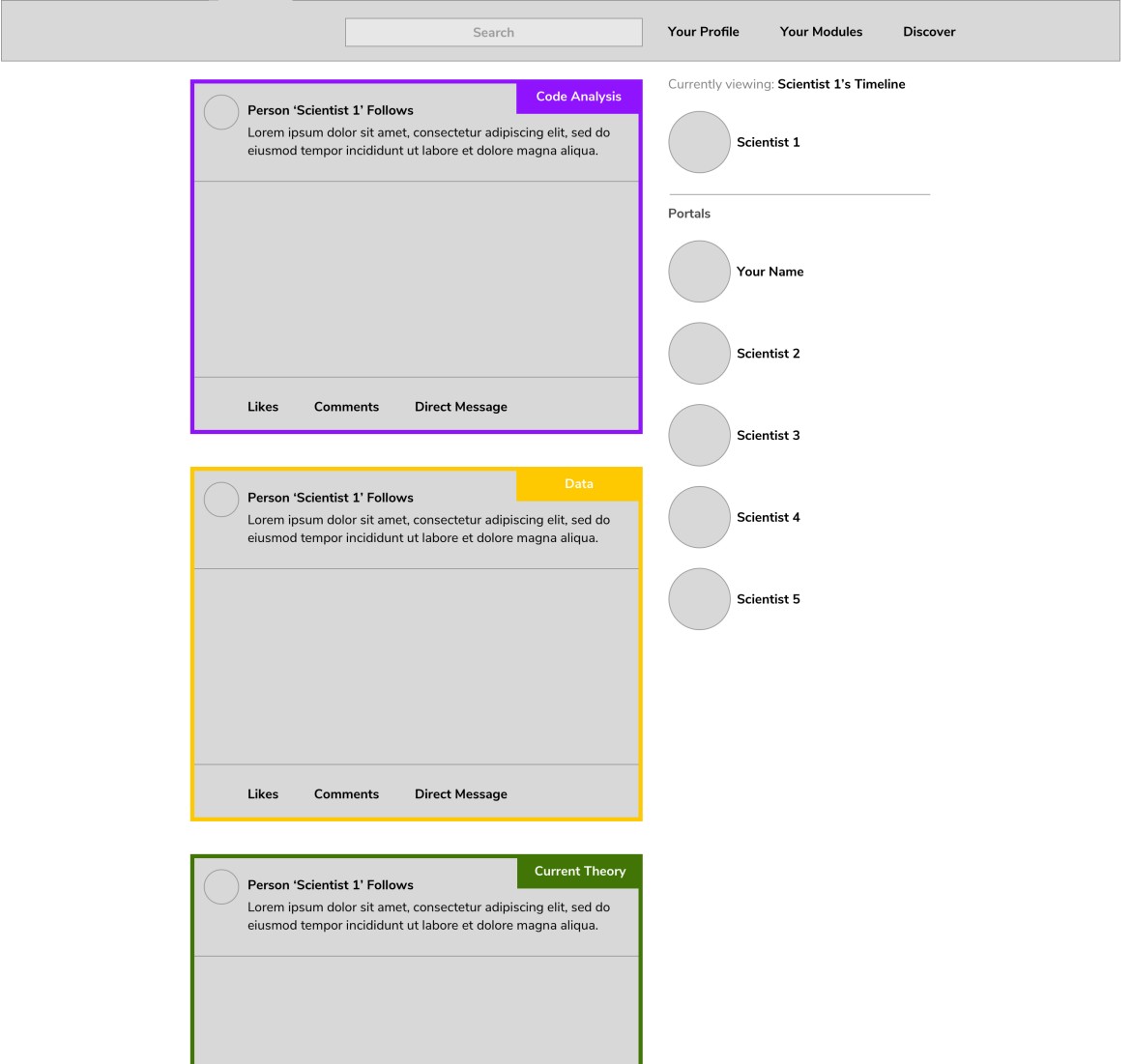

**Figure 7.** Mockup design of an additional interface for the proposed scholarly communication infrastructure. Made by Rebecca Lam, reused under CC-BY 4.0 license.

## 4. Discussion

The proposed design for decentralized, verified, provenance-based modular communication on the Dat protocol fulfills a wide conceptualization of the functions of a scholarly communications system from library and information sciences [1,2]. Due to more modular and continuous communication, it is more difficult to selectively register results when the preceding steps have publicly been registered already. Moreover, the time of communication is decided by the researcher, making it more feasible for researchers to communicate their research efforts without biases introduced at the journal stage. Certification of results is improved by embedding the chronology of the empirical research cycle in the communication process itself, making peer-to-peer discussion constructive and less obstructed by hindsight bias [27]. Unfettered awareness of research is facilitated by using an open-by-design infrastructure that is the peer-to-peer Dat protocol. Moreover, because all content is open-by-design and independent of service platforms, text- and data-mining may be applied freely without technical restrictions by service providers. The removal of these technical and service restrictions may facilitate innovations in the discovery of content and the potential for new business models to come into existence. Based on the links between scholarly modules, the arising network structure can be used to help evaluate networks of research(ers) instead of counting publications and citations [16]. Archiving

is facilitated by making it trivially easy to create local copies of large sets of content, facilitating the Lots Of Copies Keeps Stuff Safe (LOCKSS; [28,29]) principle to be more widely used than just approved organizations. Moreover, with append-only registers, the provenance of content can also be archived more readily than it is now. These functions also apply to non-empirical research that requires provenance of information (e.g., qualitative studies).

By producing scholarly content on a decentralized infrastructure, the diversity of how research is consumed and discovered can be facilitated. Currently, content lives on the webserver of the publisher and is often solely served at the publisher's web page due to copyright restrictions (except for open access articles; [30]). If the design of the publisher's web page does not suit the user's needs (e.g., due to red color blindness affecting approximately one in 20 males and one in 100 females; [31]), there is relatively little a user can do. Moreover, service providers that are not the rightsholder (i.e., publisher) now cannot fulfill that need for users. By making all content open by default, building on content becomes easier. For example, someone can build a portal that automatically shows content with color shifting for people who have red (or other types of) color blindness. Building and upgrading automated translation services are another way of improving accessibility (e.g., translexy.com/), which is currently restricted due to copyrights. Other examples of diverse ways of consuming or discovering research might include text-based comparisons of modules to build recommender algorithms that provide contrasting and corroborating views to users (e.g., [32]). Stimulating diversity in how to consume and discover content is key to making scholarly research accessible to as many people as possible and in order to attempt to keep some pace with the tremendous amount of information published each year (>3 million articles in 2017 (https://api.crossref.org/works?filter=type:journal-article,from-pub-date: 2017,until-pub-date:2017&rows=0)). As such, we have collectively passed the point of being able to comprehend the relevant information and should no longer strive to eliminate all uncertainty in knowing, but find ways to deal with that uncertainty better [33]. As such, alternatives in consuming, discovering, and learning about knowledge are a necessity. Open Knowledge Maps is an existing example of an innovative discovery mechanism based on openly-licensed and machine-readable content [34]. There would be more smaller pieces of information in the scholarly modules approach than in the scholarly article approach, which is counterbalanced by the network structure and lack of technical restrictions to build tools to digest that information; this may make those larger amounts of smaller units (i.e., modules) more digestible than the smaller volume of larger units (i.e., articles), mitigating information onslaught [35].

The proposed design is only the first in a multi-layer infrastructure that would need to be developed moving forward. Currently, I only provide a model on the container format for how to store metadata for modules (not how the data are stored in the module itself nor how the individual could go about doing so). Moreover, how could reviews be structured to fit in such modules? As such, the next layer to the proposed infrastructure would require further specification of how content is stored. For example, for text-based modules, what file formats should be the standard or allowed? It would be unfeasible to allow any file format due to readability into the future (e.g., Word 2003 files are likely to be problematic), and issues could exacerbate if software becomes more proprietary and research uses more types of software. Standards similar to current publications could prove worthwhile for text (i.e., JATS XML), but impractical for non-technical users. As such, does the original file need to be in JATS XML when it can also easily be converted? (e.g., Markdown to JATS XML; [36]). Other specifications for data, code, and materials would also be needed moving forward (e.g., no proprietary binary files such as SPSS data files). In order to make those standards practical for individuals not privy to the technical details, the next infrastructure layer would be building user-facing applications that interface with the Dat protocol and take the requirements into account. These would then do the heavy lifting for the users, guiding them through potential conversion processes and reducing friction as much as possible. An example of a rich editing environment that takes the machine readability of scholarly text to the next level, and makes this relatively easy for the end-user, is Dokie.li (which writes to HTML; [37]). This editing environment provides a What You See Is What You Get (WYSIWYG)

editor, while at the same time providing semantic enrichments to the text (e.g., discerning between positive, negative, corroborating, or other forms of citations).

New infrastructure layers could provide a much needed upgrade to the security of scholarly communication. Many of the scholarly publisher's websites do not use an appropriate level of security in transferring information to and from the user. More specifically, only 26% of all scholarly publishers use HTTPS [38]. This means that any information transferred to or from the user can be grabbed by anyone in the physical proximity of that person (amongst other scenarios), including usernames and passwords. In other words, publisher's lack of up-to-date security practices put the user at risk, but also the publisher. Some publishers for example complained about Sci-Hub, alleging that it illegally retrieved articles by phishing researcher's credentials. A lack of HTTPS would facilitate the illegal retrieval of user credentials; hence, those publishers would ironically facilitate the kinds of activities they say are illegal [39]. Beyond the potential of missed revenue for pay-to-access publishers, security negligence is worrisome because the accuracy of scholarly content is at risk. Man-in-the-middle attacks, where a middleman inserts themselves between the user and the server, can surreptitiously distort content, with practical effects for scientific practice (e.g., changing author names) and real-life effects for professions using results for their jobs (e.g., milligram dosages replaced by gram dosages). By building a scholarly communication infrastructure on top of the Dat protocol, all communications are encrypted in transit from one end to the other by default. For the format of communications, scholarly publishers may currently be unknowing distributors of malware in their PDFs distributed to (paying) readers. More specifically, an estimated 0.3–2% of scholarly PDFs contain malware [40], although the types of malware remain ill specified. By implementing scholarly modules that are converted on the user's system (e.g., JATS XML, HTML, Markdown), the attack vector on readers of the scholarly literature can be reduced by moving away from server-side generated PDFs, which potentially contain clandestine malware.

### 4.1. Limitations

In the proposed decentralized, modular scholarly communication system, there is no requirement for scholarly profiles to be linked to their real-world entities. This means that scholarly profiles may or may not be identified. For comparison, a link to a identification is also not mandatory for ORCID identifiers. Moreover, the history of anonymous (or pseudonymous) communication has a vibrant historical context in scholarly communication (e.g., [41]) and should therefore not be excluded by the infrastructure design. However, some might view this as a limitation.

One of the major points of debate may be that the scholarly modules are chronologically ordered only (both internally and externally). As such, the temporal distance between two actions within a scholarly module or between two scholarly modules is unknown. Within a scholarly module and Dat filesystem, chronological append-only actions are more reliable to register from a technical perspective than time-based append-only registers. This has its origin in the fact that creation, modification, and last opened times can technically be altered by willing users (see for example superuser.com/questions/504829). If timestamps are altered, people can fabricate records that seem genuine and chronological, but are not, undermining the whole point of immutable append-only registers. Hardcoded timestamps in the scholarly metadata would be an even greater risk due to the potential for direct modification (i.e., it would only require editing the `scholarly-metadata.json` file in a text editor). The external ordering, that is the chronology of scholarly modules, might be gamed as well. Consider the scenario where a predictions module of Version 12 is said to be the parent of a design module of Version 26, but does not exist yet at the time of registration for the design module. An individual with malicious intentions might do this and retroactively fabricate the parent predictions. Therefore, despite a specific, persistent, and unique parent Dat link being provided, the chronology could be undermined, which in turn threatens the provenance of information. It would require some effort from said researcher to ensure subsequently that the referenced Dat link contains the postdictions, but it might be possible to fake predictions in this manner. Other mechanisms could

be put in place to verify the existence of parent links at the time of registration (which is technically feasible, but would require additional bodies of trust) or to technically investigate for filler actions in a Dat filesystems when artificially high version numbers are registered. How to game the proposed system is an active avenue for further research.

The immutability of the Dat protocol that is central to this proposal only functions when information is being shared on the network continuously. Technically, if information has not been shared yet, a user could check out an old version and create an alternative history. This could prove useful when accidental versions are registered, but could also provide incorrect provenance. When already shared, the Dat protocol rejects the content, given that it is inconsistent with previous versions. As such, as long as peers keep sharing a module once its author shares it, it is difficult to corrupt. Ongoing implementations that add a checksum to the Dat link (e.g., `dat://<hash>@<checksum>+<version>`) could help further reduce this issue.

Despite the potential of building an open-by-design scholarly infrastructure on top of the Dat protocol, there are also domains where advances need to be made. Until those advances are made, widespread use in the form of a scholarly communication system remains impractical and premature (note that no technical limitations prevent an implementation of the same modular structure on current technologies, for example GitHub). These developments can occur asynchronously of the further development of this scholarly communication infrastructure. Amongst others, these domains include technical aspects and implementations of the Dat protocol itself, implementations of APIs built on top of it, legal exploration of intellectual property on a peer-to-peer network, privacy issues due to high difficulty of removing content permanently once communicated, the usability of the proposed scholarly infrastructure, and how to store information in the modules that is machine readable, but also easy-to-use for individuals.

The Dat protocol is functional, but is currently limited to NodeJS and single-user write access. Because it is currently only available in NodeJS, the portability of the protocol is currently restricted to JavaScript environments. An experimental implementation of the Dat protocol is currently being built in Rust (https://github.com/datrs) and in C++ (https://github.com/datcxx), which would greatly improve the availability of the protocol to other environments. Moreover, by being restricted to single-user write access, Dat archives are not really portable across machines or users, although work on multi-user write (i.e., multiple devices or users) has recently been released (https://github.com/mafintosh/hyperdb). Other APIs built on top of the Dat protocol that are essential to building a proposed infrastructure, such as `webdb`, also need to be further refined in order to make them worthwhile. For example, `webdb` currently does not index versioned Dat links, but simply the most recent versions. As such, the indexing of versioned references is problematic at the moment, but can be readily tackled with further development. If these and other developments continue, the benefits of the protocol will mature, may become readily available to individuals from within their standard browser, and become more practical for collaboration. Considering this, the proposed design is imperfect, but timely, allowing for community-driven iterations into something more refined, as the implementations of the Dat protocol are also refined and may become more widely used.

Instead of logging in with passwords, the Dat protocol uses cryptographical verification using a public-private key pair. A public-private key pair is similar to the lock-key pair we know from everyday life. This also means that if the (private) key is lost, a researcher can get locked out of their profile. Similarly, if the (private) key gets stolen, it might give access to people other than the researcher. How to handle private keys securely in a user-friendly manner is an important issue in further development of this scholarly communication system. Regardless, this changes the threat model from centralized leaks (for example, of plaintext passwords by Elsevier; https://perma.cc/6J9D-ZPAW) to decentralized security. This would make the researcher more in control, but also more responsible, for his/her operational security.

Despite the Dat protocol's peer-to-peer nature, intellectual property laws still ascribe copyrights upon creation and do not allow copying of content except when explicitly permitted through

non-restrictive licenses by the authors [42]. As such, intellectual property laws could be used to hamper widespread copying when licensing is neglected by authors. Legal uncertainty here might give rise to a chilling effect to use the Dat protocol to share scholarly information. Moreover, it seems virtually impossible to issue takedown notices for (retroactively-deemed) illicit content on the Dat protocol without removing all peer copies on the network. As a result of this, social perception of the Dat protocol might turn negative if high-profile cases of illicit or illegal sharing occur (regardless of whether that is scholarly information or something else). However, just as the web requires local copies in cache to function and which lawmakers made legal relatively quickly when the web was becoming widespread, the wider implementation of peer-to-peer protocols to share content might also require reforms to allow for more permissive copying of original content shared on the network. Regardless, legal issues need to be thought about beforehand, and users should be made aware that they carry responsibility for their shared content. Given its inherent open and unrestricted sharing design, it would make sense to use non-restrictive licenses on the scholarly modules by default to prevent these legal issues for researchers wanting to reuse and build on scholarly modules.

Similarly, we need to take seriously the issue that information on the network, once copied by a peer or multiple peers, is increasingly unlikely to be uncommunicated. The implications of this in light of privacy legislation, ethical ramifications, and general negative effects should not be underestimated. Because a Dat filesystem has a stable public key and stores versions, the content remains available even if the content is deleted from the filesystem. That is, users could go to an older version and still find the file that was deleted. The only way to truly undo the availability of that information is to remove all existing copies. Hence, it is worthwhile to ask the question whether scholarly research that is based on personal data should ever be conducted on the individual level data or whether this should be done on higher level summaries of relations between variables (e.g., covariance matrices). How these summaries can be verified would remain an issue to tackle. Conversely, the limitation with respect to privacy is also a benefit with regards to censorship, where information would also be much harder to censure (in stark contrast to publishers that might be pressured by governments; [43]). Moreover, we might start thinking about the ownership of data in research. In the case of human subject research, researchers now collect data and store them, but we might consider decentralized data collection where human participants produce their own data locally and simply permit a researcher to ingest that into an analysis process (creating throwaway databases themselves with `webdb` for example). This would in turn return ownership to the participant and benefit the transparency of data generated.

Bandwidth and persistent peers on the Dat protocol are highly correlated issues that are key to a usable decentralized infrastructure. When there are few peers on the network, information redundancy is low, content attrition is (potentially) high, and bandwidth will be limited. Subsequently, a maximum data transfer of 40 KB/s may be possible when few peers with restricted bandwidth are available and are farther removed on the physical network. Vice versa, in the most optimal scenario, data transfer could reach the maximum of the infrastructure between peers (e.g., 1 GB/s on peers located on an intranet). Considering that replicating Dat filesystems is relatively easy given storage space, it could be done by individuals, and (university) libraries seem particularly qualified and motivated candidates for persistent hosting of content on the Dat network. These organizations often have substantial server infrastructure available, would facilitate high data transfer speeds, and also have a vested interested in preserving scholarly content. With over 400 research libraries in Europe and over 900 academic libraries in Africa (http://db.aflia.net/list/?q=6&m=n) alone, the bandwidth and redundancy of scholarly content could be addressed if sufficient libraries participate in rehosting content. Moreover, the peer-to-peer nature would also allow for researchers to keep accessing content in the same way when the content is rehosted on the intranet and the wider connection has service interruptions.

## 4.2. Conclusions

This semi-technical proposal for verified, modular, and provenance-based scholarly infrastructure on the Dat protocol synthesized meta-research, technical developments of new web protocols, real-life

issues in a lack of diversity for consuming scholarly research, and library and information sciences' perspectives on the five functions scholarly communication is supposed to fulfill. With this initial proposal, a scholarly commons seems feasible. The proposal provides a more complete and less biased register of information than the current article-based system. Moreover, it facilitates more constructive certification discussions and allows anyone with access to the Internet to participate. It also provides archival support of the distribution, which anyone may meaningfully contribute to if they have the physical means. This proposal also may provide new ways of evaluating, consuming, and discovering research. The decentralized nature of the Dat protocol requires less trust to be put in institutions to maintain key data stores that are the foundation of any infrastructure and replaces it with widespread distribution of that information. However, technological, legal, and social developments need to occur asynchronously to make this a reality.

**Supplementary Materials:** File S1. Overview of the original Dat links corresponding to shortened links: https://github.com/chartgerink/2018dat-com/raw/master/assets/mock-modules-overview.ods.

**Funding:** This research was partly funded by the Mozilla Foundation.

**Acknowledgments:** The author thanks everyone who has helped incubate this idea in the discussions over the last two years.

**Conflicts of Interest:** The author declares no conflict of interest.

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
