# Peer review of "Verified, Shared, Modular, and Provenance Based Research Communication with the Dat Protocol"

_publications, doi:10.3390/publications7020040_

Round 1

Reviewer 1 Report

NOTE: the comments box in the online submission system is removing my formatting. Hence, I've attached the markdown source for this review.

Hartgerink proposes a loosely-defined protocol for scholarly communication based on Dat. Dat provides peer-to-peer storage organized into filesystems, which are self-contained directories that are sort of like Git repositories. However, unlike Git repositories, the address of a Dat filesystem provides a persistent ID to identify the filesystem as well as to verify write operations.

The manuscript proposes organizing scholarly research into modules, each stored by a separate Dat filesystem. In addition to modules, scholars would create filesystems to represent themselves referred to as profiles. Since Dat filesystems can contain anything, the proposed system depends on scholars adhering to specific standards, such that their modules/profiles are interpretable by other participants. This is similar to HTML pages are free to put anything in their <head>, but certain elements are interpreted in special ways according to standards. One benefit of this approach is that no central authority is required. One downside is that not all modules will encode information in consistent ways.

It can be a bit challenging to visualize how scholarship can be organized into modules. However, I think Git repositories are a good example with substantial precedent. If scholars become accustomed to making a repository for every project, as many currently do, how to structure scholarship in modules may become more widely understood. One aspect missing from the manuscript is comparing the proposed ecosystem to one where every research uses public Git repositories to create the same profile/module structure, but without using Dat.

The manuscript provides a thought-provoking proposal on how a distributed scientific ecosystem could store its information. As such, I found it to be a valuable contribution to the field of open science. Nonetheless, such a system is far off and several obstacles remain. For example, if a scholar loses their private key, they can no longer edit their module/profile. Alternatively, if their private key is leaked, anyone can edit their module/profile. Furthermore, it is challenging to assign a real identity to a digital identity. The current proposal doesn't seem to contain any mechanisms to verify that scholars are who they claim. Other issues may arise like plagiarism. For example, a researcher could monitor the network for new modules and immediately copy them, but assigning their authorship. Note that plagiarism is not unique to the proposed system, but may become more difficult to address if their are no trusted intermediaries.

Immutability

The manuscript assumes that Dat archives are immutable in many places. In the abstract:

All these scholarly modules would be communicated on the new peer-to-peer Web protocol Dat (datproject.org), which provides a decentralized register that is immutable

In the "Dat protocol" section:

The persistent public key combined with the append-only register, results in persistent versioned addresses for filesystems that also ensure content integrity. … By appending +5 to the public key (dat://0c66...613+5) we can view the Dat filesystem as it existed at version 5 and be ensured that the contents we receive are the exact contents at that version.

However, according to my understanding, Dat archives have no mechanism to ensure immutability. Anyone who possesses the private key to a Dat can create multiple divergent histories and there is no protocol-level mechanism for reaching consensus over which history is correct.

Content addressing would be one solution to ensure integrity when referencing a specific revision. hypercore-strong-linkmay be one implementation of this. Another implementation would be for the modules property of a scholar's Dat to specify a content checksum in addition to a revision number.

Content addressing protects against links resolving to a modified revision. However, it does not fix the underlying problem that history can be rewritten. Timestamps, such as those implemented in OpenTimestamps, could anchor Dats to an more-immutable & timestamped ledger like Bitcoin. If scholarly Dats were only recognized if they contained valid timestamps, retroactively editing revision history would become infeasible. Alternatively, perhaps scholarly institutions could be trusted to monitor for rewritten Dat histories and apply tools like OpenTimestamps in bulk to all known scholarly Dats.

The Merkle Tree figure seems a bit under-explained. Do data blocks L1, L2, L3, L4 correspond to file1, file2, file3, file4? It wasn't clear to me how a put or del operation would be applied to the Merkle Tree. I found this alternative explanation of Dat's Merkle Tree usage helpful.

Dat availability

Would it make sense to use an existing Dat-to-HTTP gateway to allow users without Beaker to view the Dat archives? For example, this link provides HTTP access to the "Summary" Dat.

When I attempt to view the "Prototype" dat shown in Figure 5 (dat://b068f5365f26491557dce8da1fe2f60ec5bda681424970673059228811b193dd), Beaker Browser returned the error message that "It doesn't seem like anybody is sharing this site right now." It is important, of course, that all the the Dats referenced in this manuscript are persistently available. How are the Dats currently hosted?

The discussion mentions that university libraries are appropriate agents for persistent hosting of Dats. I agree, but is there an immediate solution to maintaining availability of a Dat? While I agree that in the longterm, the Dat architecture could be more robust and persistent than the current centralized web, the unavailability of some example Dats shows this is currently not the case.

Cell I29 in the mock-modules-overview.ods has text but should probably be blank.

Intermediaries

The manuscript compares its proposed system to existing scholarly profiles, stating:

A decentralized scholarly profile in a Dat filesystem is similar and provides a unique ID (i.e., public key) for each researcher. However, researchers can modify their profiles freely because they retain full ownership and control of their data (as opposed to centralized profiles) and are not tied to one platform.

However, this assumes scholars would run their own Dat infrastructure, which seems somewhat unlikely (at least for the vast majority of scholars). More likely in my opinion is that scholars will rely on various interfaces and intermediaries that abstract away the underlying interaction with the Dat archive. While these services could be designed in such a way where scholars retain full control (i.e. possess their private key and perform all signatures client-side), services could also be custodial. If cryptocurrency wallets and storage are any lesson here, it's evident that many users will relinquish ownership of private keys for convenience. One possibility is that custodial entities may arise that offer proprietary platforms for interacting with Dat archives. I can imagine certain publishers being keenly interested in applying the gatekeeper / subscription model to future systems.

Typos

"regarded as actions [to] append to a register"

"main file for Alice her profile"

"More specifically, Alice her own profile"

Author Response

Hartgerink proposes a loosely-defined protocol for scholarly communication based on Dat. Dat provides peer-to-peer storage organized into filesystems, which are self-contained directories that are sort of like Git repositories. However, unlike Git repositories, the address of a Dat filesystem provides a persistent ID to identify the filesystem as well as to verify write operations.

The manuscript proposes organizing scholarly research into modules, each stored by a separate Dat filesystem. In addition to modules, scholars would create filesystems to represent themselves referred to as profiles. Since Dat filesystems can contain anything, the proposed system depends on scholars adhering to specific standards, such that their modules/profiles are interpretable by other participants. This is similar to HTML pages are free to put anything in their <head>, but certain elements are interpreted in special ways according to standards. One benefit of this approach is that no central authority is required. One downside is that not all modules will encode information in consistent ways.

It can be a bit challenging to visualize how scholarship can be organized into modules.

1.1 I agree. I added a figure from a similar project, Octopus, in the fourth paragraph to better depict these steps.

However, I think Git repositories are a good example with substantial precedent. If scholars become accustomed to making a repository for every project, as many currently do, how to structure scholarship in modules may become more widely understood.

1.2 The reviewer provides an interesting analogy. Git repositories are typically used to store entire research cycles instead of research steps, making this analogy incomplete for this specific concept and potentially confusing more than clarifying. I hope the addition in 1.1 will help clarify this.

One aspect missing from the manuscript is comparing the proposed ecosystem to one where every research uses public Git repositories to create the same profile/module structure, but without using Dat.

1.3 See 1.2 why this is omitted. Nonetheless, I added a note in the limitations which reads as follows

note that no technical limitations prevent an implementation of the same modular structure on current technologies, for example GitHub

The manuscript provides a thought-provoking proposal on how a distributed scientific ecosystem could store its information. As such, I found it to be a valuable contribution to the field of open science.

1.4 Thank you for the affirming words :)

Nonetheless, such a system is far off and several obstacles remain.

1.5 I agree with the reviewer and see these challenges as key points moving forward.

For example, if a scholar loses their private key, they can no longer edit their module/profile. Alternatively, if their private key is leaked, anyone can edit their module/profile.

1.6 A valid point. In the limitations section I added a paragraph on this to include more recent developments

Instead of logging in with passwords, the Dat protocol uses cryptographical verification using a public-private key pair. A public-private key pair is similar to the lock-key pair we know from everyday life. This also means that if the (private) key is lost, a researcher can get locked out from their profile. Similarly, if the (private) key gets stolen, it might give access to people other than the researcher. How to securely handle private keys in a user-friendly manner is an important issue in further development of this scholarly communication system. Regardless, this changes the threat model from centralized leaks (for example of plaintext passwords by Elsevier; https://perma.cc/6J9D-ZPAW) decentralized security. This would make the researcher more in control, but also more responsible, for their operational security.

Furthermore, it is challenging to assign a real identity to a digital identity. The current proposal doesn't seem to contain any mechanisms to verify that scholars are who they claim.

1.7 Public keys serve as direct verification of the person one is interacting with. For example, I meet the reviewer at a conference, who gives me a QR code of their 64-character public key. I open this, and that way I am sure of the identity being theirs (as sure as I can be at least).

The other way around, when a researcher stumbles upon a public key in the network, it all depends on the chain of reference: Do I trust the intermediaries in the network who supply me the public key of the individual I do not know directly?

Moreover, not all researcher profiles need to be assigned an identity that is available in the "real world". This allows for anonymous users to be created, which is an important part of scholarly communication when repercussions might be a result of the results.

In sum, although I agree with the reviewer's point that assigning a real identity is challenging, it is part of the design to allow for this. However, I agree I did not extend upon this in the paper and added a paragraph in the limitations section that reads as follows

In the proposed decentralized, modular scholarly communication system there is no requirement for scholarly profiles to be linked to their real-world entities. This means that scholarly profiles may or may not be identified. For comparison, a link to a identification is also not mandatory for ORCID identifiers. Moreover, the history of anonymous (or pseudonymous) communication has a vibrant historical context in scholarly communication  and should therefore not be excluded by the infrastructure design. However, some might view this as a limitation.

Other issues may arise like plagiarism. For example, a researcher could monitor the network for new modules and immediately copy them, but assigning their authorship. Note that plagiarism is not unique to the proposed system, but may become more difficult to address if their are no trusted intermediaries.

1.8 A valid point. At an individual level, the public keys of individuals provide the intermediary that the trust can be based on (or not). Systemically, plagiarism detection could be offered as a service (as it is now), which continually scans for posted modules from a set of users and compares it to other modules. Because all content is publicly available, building automated plagiarism scanners on top of the content is possible. Disputes about who had precedence will remain.

Immutability

The manuscript assumes that Dat archives are immutable in many places. In the abstract:

All these scholarly modules would be communicated on the new peer-to-peer Web protocol Dat (datproject.org), which provides a decentralized register that is immutable

In the "Dat protocol" section:

The persistent public key combined with the append-only register, results in persistent versioned addresses for filesystems that also ensure content integrity. … By appending +5 to the public key (dat://0c66...613+5) we can view the Dat filesystem as it existed at version 5 and be ensured that the contents we receive are the exact contents at that version.

However, according to my understanding, Dat archives have no mechanism to ensure immutability. Anyone who possesses the private key to a Dat can create multiple divergent histories and there is no protocol-level mechanism for reaching consensus over which history is correct.

1.9 The reviewer raises a strong point.

The technical discussions I'd had with the Dat team seemed to previously indicate reverting these older versions was not possible and the documentation supported it (where checking out an older version resulted in read-only capabilities).

Recent discussions seem to indicate otherwise, partly. I've added a paragraph in the discussion to highlight this

The immutability of the Dat protocol that is central to this proposal only functions when information is being shared on the network continuously. Technically, if information has not been shared yet, a user could check out an old version and create an alternative history. This could prove useful when accidental versions are registered, but could also provide incorrect provenance. When already shared, the Dat protocol rejects the content given that it is inconsistent with previous versions. As such, as long as peers keep sharing a module once its author shares it, it is difficult to corrupt. Ongoing implementations that add a checksum to the Dat link (e.g., dat://<hash>@<checksum>+<version>) could help further reduce this issue.

Content addressing would be one solution to ensure integrity when referencing a specific revision. hypercore-strong-link may be one implementation of this. Another implementation would be for the modules property of a scholar's Dat to specify a content checksum in addition to a revision number.

1.10 Strong links are a development I am looking into, and have mentioned in 1.9.

Content addressing protects against links resolving to a modified revision. However, it does not fix the underlying problem that history can be rewritten.

1.11 See also 1.9. I do agree with the reviewer that I should look into how to corrupt the provenance, which I've added to my list for further work.

Timestamps, such as those implemented in OpenTimestamps, could anchor Dats to an more-immutable & timestamped ledger like Bitcoin. If scholarly Dats were only recognized if they contained valid timestamps, retroactively editing revision history would become infeasible. Alternatively, perhaps scholarly institutions could be trusted to monitor for rewritten Dat histories and apply tools like OpenTimestamps in bulk to all known scholarly Dats.

1.12 See 1.9

The Merkle Tree figure seems a bit under-explained. Do data blocks L1, L2, L3, L4 correspond to file1, file2, file3, file4? It wasn't clear to me how a put or del operation would be applied to the Merkle Tree. I found this alternative explanation of Dat's Merkle Tree usage helpful.

1.13 The figure is unrelated to the actions and was mainly used to conceptually depict the Merkle Tree. I added a clarification in the figure caption to differentiate the two ("These do not correspond to the actions outlined in the text.")

Dat availability

Would it make sense to use an existing Dat-to-HTTP gateway to allow users without Beaker to view the Dat archives? For example, this link provides HTTP access to the "Summary" Dat.

1.14 I do not include a gateway because the APIs that the prototype uses are specific to Beaker and would not function for users without Beaker. Note this is a proof of concept and not something to actually use.

When I attempt to view the "Prototype" dat shown in Figure 5 (dat://b068f5365f26491557dce8da1fe2f60ec5bda681424970673059228811b193dd), Beaker Browser returned the error message that "It doesn't seem like anybody is sharing this site right now." It is important, of course, that all the the Dats referenced in this manuscript are persistently available. How are the Dats currently hosted?

1.15 The Dats are currently hosted through Hashbase.io; (semi-)persistent hosting was what I intended to do. I checked all Dats referenced in the manuscript and it was only the Prototype that no longer functioned. Given that the main code is available in the manuscript and I no longer had the exact version, I removed the references to the Prototype link and restricted the text to describing it (it is outdated now anyway).

The discussion mentions that university libraries are appropriate agents for persistent hosting of Dats. I agree, but is there an immediate solution to maintaining availability of a Dat? While I agree that in the longterm, the Dat architecture could be more robust and persistent than the current centralized web, the unavailability of some example Dats shows this is currently not the case.

1.16 As the reviewer mentioned themselves, this is a design that is some time out from wide adoption. I find it interesting that there should be an immediate solution to the issue of rehosting. But yes, there are immediate solutions available for users (as hashbase.io is built for example) or library solutions that can be built rather easily by creating a daemon that synchronizes all the Dats stored in a specific database. These could be run as relatively low overhead Docker containers.

Cell I29 in the mock-modules-overview.ods has text but should probably be blank.

1.17 Correct and updated.

Intermediaries

The manuscript compares its proposed system to existing scholarly profiles, stating:

A decentralized scholarly profile in a Dat filesystem is similar and provides a unique ID (i.e., public key) for each researcher. However, researchers can modify their profiles freely because they retain full ownership and control of their data (as opposed to centralized profiles) and are not tied to one platform.

However, this assumes scholars would run their own Dat infrastructure, which seems somewhat unlikely (at least for the vast majority of scholars). More likely in my opinion is that scholars will rely on various interfaces and intermediaries that abstract away the underlying interaction with the Dat archive. While these services could be designed in such a way where scholars retain full control (i.e. possess their private key and perform all signatures client-side), services could also be custodial. If cryptocurrency wallets and storage are any lesson here, it's evident that many users will relinquish ownership of private keys for convenience. One possibility is that custodial entities may arise that offer proprietary platforms for interacting with Dat archives. I can imagine certain publishers being keenly interested in applying the gatekeeper / subscription model to future systems.

1.18 I do not assume anyone to run their own Dat seeders, which is exactly why I see a role for (university) libraries as the reviewer acknowledged in 1.16. I agree custodial services may be built, but I find it a potential false equivalency to compare them to crypto wallets, which in part was stimulated by the exchange market aspect of it (e.g., coinbase). Nonetheless, I agree with the reviewer the convenience aspect is important, and is something I have not thought enough about. I will in further development of this idea :)

Typos

"regarded as actions [to] append to a register"

1.19 Updated

"main file for Alice her profile"

1.20 Updated to Alice's

"More specifically, Alice her own profile"

1.21 Updated to "Alice's personal profile"

Reviewer 2 Report

In this paper, the author proposes a provenance-based system of scholarly communication. This scholarly infrastructure facilitates the more efficient and complete register of scholarly information that is not restricted to “officially published” scientific works. Essentially there are two types of entities in the network: researchers, and scholarly information/products. The "scholarly profiles" represent information of scholars, and the "scholarly modules" represent information of a variety of scholarly works. They form a decentralized network of scholarly information, with versioning, and is inherently open to the world.

1.      Although this system can potentially facilitate more constructive discussions around certification (which is equally if not more important than the “seals of approval”), it is insufficient to claim that this system can fulfill the function of certification. The certification process requires much more than making the work openly available. Peer review is flawed in many ways, but it is still so far the most effective way to control the quality (gate-keeping) of science. The author needs to provide more discussions on this.

2.      Building upon the first point, the process of certification also functions as an effective instrument in the “natural selection” of science. No one can read everything. How to make the “better” science more discoverable in the Dat system? I’d suggest the author elaborate more on this point as well.

3.      In the current scholarly communications system, there are multiple types of scholarly publications, among which journal article is a major type. However, there are also monographs (prevalent in the humanities), conference papers (e.g., in computer science and related disciplines is a more acknowledged and accepted practice), not to mention the publication and self-archiving of research data. Please include these as background.

4.      The author needs to provide more discussion on the role and flaws of preprints (highly relevant here).

5.      Relevant to the third point, the terms article-based system” and "journal/article based system" (Line 191) have been used to describe “the current” scholarly communications system. It’d be better to use one term consistently (and provide a rationale for using it). This would be helpful to ground the “module-based” conception, which is an essential part of this manuscript.  

6.      Line 184: “In a typical theory-testing research study” --> what for the non-theory-testing research? The rationale (theoretical support & empirical evidence) behind the division/categorization of these modules needs to be better described.

7.      Line 209: "if the module is a direct consequence of a previous registered module" more details need to be provided here in terms of what a "consequence" can be. Can you provide some examples? Can these Dat links indicate relationships other than “consequence”?

8.      The author briefly mentioned the detection of filler actions at the end. It’d be better to see more discussions addressing the issue of gaming, particularly when incentivizing is also a function in scholarly communication that Dat would like to fulfill.

Overall, great work. I would recommend this paper to be published after these points have been addressed. Thank you.

Author Response

In this paper, the author proposes a provenance-based system of scholarly communication. This scholarly infrastructure facilitates the more efficient and complete register of scholarly information that is not restricted to “officially published” scientific works. Essentially there are two types of entities in the network: researchers, and scholarly information/products. The "scholarly profiles" represent information of scholars, and the "scholarly modules" represent information of a variety of scholarly works. They form a decentralized network of scholarly information, with versioning, and is inherently open to the world.

Although this system can potentially facilitate more constructive discussions around certification (which is equally if not more important than the “seals of approval”), it is insufficient to claim that this system can fulfill the function of certification. The certification process requires much more than making the work openly available. Peer review is flawed in many ways, but it is still so far the most effective way to control the quality (gate-keeping) of science. The author needs to provide more discussions on this.

2.1 I agree with the reviewer on this point. In order to clarify I changed the description to "certification by peer review is supplemented by embedding chronology to prevent misrepresentation [...]"

Building upon the first point, the process of certification also functions as an effective instrument in the “natural selection” of science. No one can read everything. How to make the “better” science more discoverable in the Dat system? I’d suggest the author elaborate more on this point as well.

2.2 Very good point of the reviewer regarding discoverability in the proposed system. Given the current 'natural selection of bad science' (Smaldino & McElreath, 2016) I would argue the current system is failing at being an effective instrument to filter out "worse" science. Moreover, the discussion of the reviewed manuscript already discussed discoverability at length in the discussion, concluded with "There would be more smaller pieces of information in the scholarly modules approach, which is counterbalanced by the network structure and lack of technical restrictions to build tools to digest that information --- this may make those larger amounts of smaller units (i.e., modules) more digestible than the smaller volume of larger units (i.e., articles)"

In the current scholarly communications system, there are multiple types of scholarly publications, among which journal article is a major type. However, there are also monographs (prevalent in the humanities), conference papers (e.g., in computer science and related disciplines is a more acknowledged and accepted practice), not to mention the publication and self-archiving of research data. Please include these as background.

2.3 The reviewer has a valid point, but I refrain from including these because I am talking about the main form of communication as it stands right now.

The author needs to provide more discussion on the role and flaws of preprints (highly relevant here).

2.4 The reviewer seems to imply that there are issues with certification in preprints, which would be apt to discuss in a paper about preprints. It seems like ill-suited for the current paper, however, given that preprints are not at all the subject of the manuscript nor are they mentioned anywhere in the text.

Relevant to the third point, the terms “article-based system” and "journal/article based system" (Line 191) have been used to describe “the current” scholarly communications system. It’d be better to use one term consistently (and provide a rationale for using it). This would be helpful to ground the “module-based” conception, which is an essential part of this manuscript.

2.5 Removed the singular occurrence of "journal/article based"

Line 184: “In a typical theory-testing research study” --> what for the non-theory-testing research? The rationale (theoretical support & empirical evidence) behind the division/categorization of these modules needs to be better described.

2.6 Inserted a clarification ""

Line 209: "if the module is a direct consequence of a previous registered module" more details need to be provided here in terms of what a "consequence" can be. Can you provide some examples? Can these Dat links indicate relationships other than “consequence”?

2.7 Adjusted to "following step" for clarity.

The author briefly mentioned the detection of filler actions at the end. It’d be better to see more discussions addressing the issue of gaming, particularly when incentivizing is also a function in scholarly communication that Dat would like to fulfill.

2.8 See also 1.9. The initial proposal for incentives and evaluation was done in doi:10.3390/publications6020021.

Overall, great work. I would recommend this paper to be published after these points have been addressed. Thank you.

2.9 Thank you for the kind words :)